# Inhibition of the IL-18 Receptor Signaling Pathway Ameliorates Disease in a Murine Model of Rheumatoid Arthritis

**DOI:** 10.3390/cells9010011

**Published:** 2019-12-18

**Authors:** Yuji Nozaki, Jinhai Ri, Kenji Sakai, Kaoru Niki, Koji Kinoshita, Masanori Funauchi, Itaru Matsumura

**Affiliations:** Department of Hematology and Rheumatology, Kindai University Faculty of Medicine; Osaka-sayama, Osaka 589-8511, Japan; jinhai@med.kindai.ac.jp (J.R.); kenji-s-101@med.kindai.ac.jp (K.S.); komyodai9300@yahoo.co.jp (K.N.); kkino@med.kindai.ac.jp (K.K.); mn-funa@med.kindai.ac.jp (M.F.); nozaki0516@yahoo.co.jp (I.M.)

**Keywords:** rheumatoid arthritis, cytokine, IL-18, SOCS, murine

## Abstract

Interleukin (IL)-18 expression in synovial tissue correlates with the severity of joint inflammation and the levels of pro-inflammatory cytokines. However, the role of the IL-18/IL-18 receptor-alpha (Rα) signaling pathway in autoimmune arthritis is unknown. Wild-type (WT) and IL-18Rα knockout (KO) mice were immunized with bovine type II collagen before the onset of arthritis induced by lipopolysaccharide injection. Disease activity was evaluated by semiquantitative scoring and histologic assessment. Serum inflammatory cytokine and anticollagen antibody levels were quantified by an enzyme-linked immunosorbent assay. Joint cytokine and matrix metalloproteinases-3 levels were determined by a quantitative polymerase chain reaction. Splenic suppressors of cytokine signaling (SOCS) were determined by Western blot analysis as indices of systemic immunoresponse. IL-18Rα KO mice showed lower arthritis and histological scores in bone erosion and synovitis due to reductions in the infiltration of CD4+ T cells and F4/80+ cells and decreased serum IL-6, -18, TNF, and IFN-γ levels. The mRNA expression and protein levels of SOCS3 were significantly increased in the IL-18Rα KO mice. By an up-regulation of SOCS, pro-inflammatory cytokines were decreased through the IL-18/IL-18Rα signaling pathway. These results suggest that inhibitors of the IL-18/IL-18Rα signaling pathway could become new therapeutic agents for rheumatoid arthritis.

## 1. Introduction

Interleukin (IL)-18 is a proinflammatory cytokine produced by antigen-presenting cells (APCs) and T cells such as macrophages, dendritic cells (DCs) and CD4+ T cells. The IL-18 receptor (IL-18R) is a heterodimer comprised of a signaling IL-18Rβ subunit (also called IL-1RAcPL and IL-1R7) and a ligand-binding IL-18Rα subunit. Downstream from IL-18R, the signaling activates interleukin-1 receptor-associated kinase 4 (IRAK4) and the adaptor molecule MyD88, in a scenario similar to that of other IL-1 and Toll-like receptors (TLRs) [1,2,3]. Evidence has been presented that IL-18 plays a prominent role in the onset and maintenance of an inflammatory response during rheumatoid arthritis. It was reported that IL-18 expression in synovial tissue correlates with the severity of joint inflammation and the levels of pro-inflammatory cytokines tumor necrosis factor alpha (TNFα) and IL-1β [4].

Several studies have indicated that IL-18 may play an important role in the pathophysiology of collagen-induced arthritis (CIA), which is a well-established autoimmune animal model of human rheumatoid arthritis [5]. There are several reports that a local neutralization of IL-18 by the naturally occurring IL-18 binding protein (IL-18BP) or a systemic neutralization by specific antibodies results in the amelioration of CIA, associated with reduced inflammation and cartilage erosion [6,7,8].

The inhibition of cytokine signaling by the suppressors of cytokine signaling (SOCS) family constitutes a major negative feedback mechanism to prevent runaway inflammation. The transcription of SOCS proteins is rapidly upregulated in cells stimulated with cytokines. The SOCS then reduce the impact of cytokines by interacting with Janus kinases (JAKs) and by other mechanisms [9]. The roles of SOCS1 and −3 are underscored by the higher levels of the proinflammatory p38α and p38β mitogen-activated protein kinases (MAPKs) observed in IL-18Rα-deficient mouse embryonic fibroblasts (MEFs) compared to wild-type (WT) and IL-18-deficient MEFs. Since both SOCS1 [10,11,12] and SOCS3 [13,14] interfere with the IL-1/TLR-MAPK/nuclear factor-kappa (NF-κ)B pathway signaling, the increase in activity of these kinases in IL-18Rα-deficient cells may be due to a lesser degree of inhibition resulting from the reduced levels of both of these SOCS family members.

In the present study, we confirmed that IL-18Rα was aberrantly expressed in the lymph nodes (LN) and splenocytes of mice with lipopolysaccharide (LPS)-induced arthritis. We hypothesized that if arthritis becomes hyper-responsive to LPS, which is a component of Gram-negative bacteria, this hyper-responsiveness may increase the levels of circulating inflammatory cytokines such as TNF, interferon-gamma (IFN-γ), and IL-6 and result in the blocking of the IL-18Rα-mediated signaling pathway by SOCS, and ultimately an amelioration of the LPS-induced arthritis. The results of our experiments demonstrated that the blockade of the IL-18Rα signaling pathway inhibited not only the proliferation of autoreactive T cells, but also the suppression of Th1 cytokines and infiltrations of inflammatory cells such as CD4+ T cells and APCs. Our results indicate that blockade of the IL-18Rα signaling pathway might become a new therapeutic strategy for RA.

## 2. Materials and Methods

### 2.1. Animals

IL-18Rα knock-out (KO) (IL-1Rrp−/−) C57BL/6 mice were kindly provided by Dr. Shizuo Akira (Osaka University, Osaka, Japan). The DBA1/J mice used as a WT control were purchased from Shizuoka Laboratory Animal Centre (Shizuoka, Japan). All mice were maintained in our specific pathogen-free animal facility. We constructed an IL-18Rα KO DBA1/J strain using a backcross-intercross breeding scheme [15]. DBA1/J mice were mated with IL-18Rα KO mice on the C57BL/6 background to yield heterozygous F1 offspring. We intercrossed F1 mice and screened the progeny by performing a polymerase chain reaction (PCR) amplification of tail genomic DNA for the mutation and IL-18Rα using specific primers [16]. These mice were backcrossed onto the DBA/1J background for seven generations. The animal protocols were approved by the Kindai University Animal Care Committee and were performed in accordance with the Kindai University Animal Care Guidelines.

### 2.2. Murine Model of Collagen-Induced Arthritis

CIA was induced according to an established method as described previously [17]. Briefly, 6-week-old male mice were immunized by a single subcutaneous injection at the base of the tail with 200 μg of bovine type II collagen (Chondrex, Seattle, WA, USA) in 0.05 M acetic acid, emulsified with an equal volume of complete Freund’s adjuvant, containing 200 μg of H37RA Mycobacterium tuberculosis (Chondrex). In all experiments, the mice were injected intraperitoneally with 50 μg/kg LPS (Escherichia coli O111:B4; Sigma-Aldrich, St. Louis, MO, USA). Blood was collected in heparinized tubules for the measurement of TNF, IFN-γ, IL-6, IL-18, and matrix metalloproteinase-3 (MMP-3). Mice in the WT and IL-18Rα KO groups were sacrificed by carbon dioxide asphyxiation at day 0 (n = 4 and 4), day 2 (n = 12 and 8), day 4 (n = 10 and 15), and day 14 (n = 8 and 6) after the LPS injection with the collection of blood as described above as well as the synovium and spleen tissue. The resulting lethality was monitored for 14 days after the LPS injection (Figure 1).

The development of CIA was assessed 2 times per week, and inflammation in the four paws of each mouse was graded from 0 to 4 as follows: grade 0, normal; grade 1, swelling and/or redness in one joint; grade 2, swelling and/or redness in a wrist joint; grade 3, swelling and/or redness in the entire paw; grade 4, deformity and/or ankylosis. The scores for each paw were summed so that the maximum possible score per mouse was 16. The paw inflammation grading was performed by two researchers using coded slides. Mice that received no bovine type II collagen or LPS were sacrificed as the normal controls (WT, n = 4; KO, n = 4).

### 2.3. Measurement of Spleen Weight and the Proliferation Assay of Splenocytes

On days 0, 2, 4, and 14, the spleens of mice sacrificed by carbon dioxide asphyxiation were removed and weighed. Single-cell suspensions were made in RPMI 1640 medium supplemented with 10% fetal calf serum and adjusted to 4 × 10^6^ cells/well. Spleen cells were plated in triplicate in a flat-bottomed 96-well plate and stimulated with type II collagen (50 μg/mL) or medium alone for 48 h in the presence of the plant mitogen concanavalin A (ConA, 1 μg/mL) from Sigma-Aldrich (St Louis, MO, USA) and LPS (2 μg/mL), according to the protocol determined in the preliminary experiments. Proliferation was assessed by a CellTiter 96^®^ AQueous One Solution Cell Proliferation Assay (Promega, Madison, WI). The CellTiter 96^®^ AQueous One Solution contains a water-soluble tetrazolium salt, i.e., MTT (3-(4,5-dimethylthiazol-2-yl)-5-(3-carboxymethoxyphenyl)-2-(4-sulfophenyl)-2H-tetrazolium, inner salt) that produces a water-soluble formazan dye upon bioreduction in the presence of an electron carrier, phenazine ethosulfate, thereby providing a simple and rapid measurement of the cell proliferation. The amount of formazan produced is proportional to the number of living cells. The detection sensitivity of the CellTiter 96^®^ AQueous One Solution assay is comparable to that of the MTT assay, but the former assay is simpler [18].

### 2.4. Histological Examination

On days 0, 2, 4, and 14, wrists from CIA mice were placed in 4% formaldehyde, embedded in paraffin, sectioned, and stained with hematoxylin and eosin (HE). Synovitis and erosions were separately scored from 0 to 3, where 0 = normal appearance, 1 = mild, 2 = moderate, and 3 = severe synovitis/erosion of cartilage and bone. A histopathological index was constructed by adding the scores from the evaluated joints in each animal. Immunohistochemical staining for CD4+ T cells and F4/80+ cells as the macrophages were performed on 4-μm thick formalin-fixed sections [16]. The number of positive cells was measured as the stained cell count/mm^2^ per slide, and the results are expressed as cells per high-power field (c/hpf). The primary monoclonal antibodies used were rat monoclonal antibody GK1.5 for CD4+ T cells (Pharmingen, San Diego, CA, USA) and F4/80 hybridoma culture supernatant (HB 198; American Type Culture Collection, Manassas, MD, USA).

### 2.5. Micro-Computed Tomography (Micro-CT)

Wrist joints of WT and IL-18Rα KO mice were imaged on day 14 using a DELPet µCT100 micro-computed tomography (micro-CT) system (Delbio, Taoyuan, Taiwan). The structural changes of the hind paws were visualized by micro-CT imaging on day 14 after the LPS injection. The mouse paws were subjected to micro-CT imaging using a LaTheta LCT-200 CT scanner (Hitachi Aloka Medical, Tokyo), and the projection images were reconstructed into three-dimensional images using VGStudio MAX software version 2.2 (Volume Graphics, Heidelberg, Germany).

### 2.6. Measurement of Anti-Collagen Antibody

Serum samples were collected by cardiac puncture on days 0, 2, 4, and 14. Each sample was analyzed in triplicate by enzyme linked immunosorbent assays (ELISAs). Pooled sera from the 8- to 10-week-old mice were used as the standard. All serum samples were diluted 1:8000 in phosphate-buffered saline (PBS). Ninety-six-well plates (CORNING, Oneonta, NY, USA) were coated with 5 μg/mL of type II collagen (Chondrex) and kept at 48 °C overnight. The plates were washed three times with PBS containing 0.05% Tween and then blocked with 3% bovine serum albumin (BSA) (Sigma-Aldrich) in PBS at 37 °C for 2 h. After the blocking solution was removed, 50 μL of 1:2000 dilutions of each serum sample in PBS was added to each of three plates and incubated at 37 °C for 45 min and then at 48 °C for 45 min. The plates were washed three times, and then 50 μL of horseradish-peroxidase-conjugated goat mouse IgG, IgG1 (PharMingen, San Diego, CA, USA), IgG2a (MP Biomedicals, Irvine, CA, USA), or IgG2b (Southern Biotech, Birmingham, AL, USA) antibodies (1:2000 dilution in PBS) was added and the plates were kept at 37 °C for 2 h. Absorbance at 450 nm of each well was measured by a microplate reader (Benchmark; Bio-Rad, Richmond, CA, USA). The antibody (Ab) titer of the samples was determined from the absorbance using a standard curve constructed for each IgG subclass. A standard curve was plotted using serum dilutions of 1:8000 (200 units/mL), 1:16,000 (100 units/mL), 1:32,000 (50 units/mL), 1:64,000 (25 units/mL), 1:128,000 (12.5 units/mL), 1:256,000 (6.25 units/mL), and 1:512,000 (3.1 units/mL).

### 2.7. Measurements of mRNA Expression in the Synovium by Real-Time PCR

We performed a real-time PCR as described previously [16] for the measurement of the mRNA expressions of TNF, GATA3, T-bet, and 18SrRNA by using FastStart DNA Master Sybr Green I (Applied Biosystems, Foster City, CA, USA) and for the measurement of IFN-γ, IL-6, -18, -18R1 (encoding IL-18Rα), SOCS3, and 18SrRNA in the synovium using TaqMan gene (Applied Biosystems). The sequences of the primer and gene database numbers are listed in Table 1; Table 2. All specific amplicons were normalized against 18SrRNA, which was amplified in the same reaction as an internal control using commercial reagents (Applied Biosystems) and is expressed as the fold-increase relative to saline-treated control mice. The relative amount of mRNA was calculated using the comparative Ct (∆∆Ct) method.

### 2.8. Fluorescence-Activated Cell Sorting (FACS) Analysis

For the assessment of intracellular cytokines by a FACSCanto II flow cytometry system (Becton Dickinson, Lincoln Park, NJ, USA), we obtained LN cells from the axillary and neck region at days 2, 4, and 14 and splenocytes at day 4 after LPS injection. LN cells and splenocytes were stained with monoclonal antibodies for IL-18R1 and intracellularly with antibodies (anti-TNFα and anti-IFN-γ) conjugated to fluorescein isothiocyanate (FITC) and phycoerythrin (PE). We assessed the intracellular cytokines by determining their percentages in CD4+ T cells, F4/80+, CD11b+, and CD11c+ cells. The flow cytometry antibodies were FITC-anti-CD4, PE-Cy7-anti-F4/80, BV421-anti-CD11b, APC-Cy7-CD11c, FITC-IL-18R1, FITC-anti-TNF, and PE-anti-IFN-γ (BD Bioscience). For all markers, an isotype-matched irrelevant monoclonal antibody was used. Cells that were fluorescing at levels above the negative control were considered positive.

### 2.9. Measurement of Cytokine Productions In Vitro

Splenocytes from the spleens of WT or IL-18Rα KO mice were stimulated with ConA (1 μg/mL) and LPS (2 μg/mL) for 24 h in the presence of type II collagen (50 μg/mL), according to the protocol established in the preliminary experiments. The concentrations of cytokines (IL-6, IL-18, IFN-γ, and TNF) in the serum and supernatant were determined using an ELISA kit for each cytokine (PharMingen).

### 2.10. Western Blotting

Proteins were extracted by homogenization of the spleen on days 2, 4, and 14 in tissue protein extraction reagent (Pierce, Rockford, IL, USA) to determine the levels of SOCS3 protein, as described previously [19]. Monoclonal anti-β-actin antibody was obtained from Santa Cruz (St. Louis, MO, USA). Anti-mouse SOCS3 antibody was obtained from Cell Signaling Technology (Danvers, MA, USA). Peroxidase-conjugated goat immunoglobulin G was purchased from Santa Cruz.

### 2.11. Statistical Analyses

Results are expressed as the mean ± standard error of the mean (SEM). Groups were compared by the unpaired *t*-test or by an analysis of variance (ANOVA) when more than two groups were compared. Probability values <0.05 were accepted as significant. The survival time was estimated using the Kaplan-Meier method. The log-rank test was used to compare survival times between groups. In vivo, the sample data were analyzed from WT and IL-18Rα KO mice; day 0 (n = 4 and n = 4), 2 (n = 12 and n = 8), 4 (n = 10 and n = 15), and 14 (n = 8 and n = 6) in all figures except the sample data in Figure 3A on day 2, 4, 14 (n = 6). In vitro, the splenocytes were extracted from the WT, IL-18Rα KO, and control mice (n = 5) in Figure 7. We analyzed the data using GraphPad Prism software version 6 (Graphpad Software, La Jolla, CA, USA).

## 3. Results

### 3.1. IL-18Rα Deficiency Improved the Arthritis Scores after CIA-Induced Arthritis

We evaluated the severity of arthritis in all four paws of IL-18Rα KO and WT mice that had received the LPS injection. As shown Figure 2A, we observed that the IL-18Rα KO mice had significantly lower arthritis scores compared to the WT mice throughout the disease course. Similarly, IL-18Rα deficiency decreased the incidence of arthritis (Figure 2B). Compared to those of the WT mice, the histological scores of the IL-18Rα KO mice were significantly lower in the wrist joints with erosions (day 4: 1.4 ± 0.3 vs. 0.4 ± 0.2; day 14: 1.8 ± 0.3 vs. 1.0 ± 0.5) and those with synovitis (day 2: 1.2 ± 0.2 vs. 0.5 ± 0.1; day 14: 2.3 ± 0.3 vs. 1.4 ± 0.3) on cartilage and bone (Figure 2C).

Figure 2D,E provides representative images of synovitis and erosions in the wrist joints and the scores from histological staining with hematoxylin and eosin (H&E) and micro-CT findings from WT and IL-18Rα KO mice with CIA on day 14. The 3D reconstruction revealed the bone erosion in the forepaws of mice from both groups. In the WT mice, the bone erosions in the area of the metatarsophalangeal joint and carpal bones revealed bone deformation and ankylosis. The IL-18Rα KO mice also exhibited bone erosion and deformation, but to a lesser degree. There were no signs of bone injury among the cartilage bones of the non-immunized mice.

### 3.2. IL-18 and IL-18Rα mRNA Expression in the Synovium After CIA-Induced Arthritis

To determine whether CIA-induced arthritis stimulates the IL-18/IL-18Rα signaling pathway, we measured the mRNA expressions of IL-18 and IL-18Rα in the synovium of WT DBA/1J after an LPS injection by real-time PCR (Figure 3A,B). Compared to the expressions in the control mice, the peaks of the IL-18 and IL-18Rα mRNA expressions were significantly higher on day 4 (unlike the values on days 2 and 14). Compared to the expressions on day 4, the IL-18 and IL-18Rα mRNA expressions on day 14 were significantly decreased as follows: IL-18, 14.7 ± 6.8 vs. 6.3 ± 5.7; IL-18Rα, 98.8 ± 68.1 vs. 20.3 ± 20.0, respectively.

We also examined the expressions of IL-18R1^+^ on CD4^+^ T cells, F4/80^+^ cells, CD11b^+^ cells, and F4/80^+^CD11b^+^ cells by performing a FACS analysis of LN cells in WT and IL-18Rα KO mice on day 4 (Figure 3C). CIA-induced arthritis resulted in increased proportions of IL-18R1^+^ on these cells in WT mice compared to the proportions in IL-18Rα KO mice: CD4^+^ T cells, 2.4 ± 0.1 vs. 0.0 ± 0.0; F4/80^+^ cells, 0.2 ± 0.0 vs. 0.0 ± 0.0; CD11b^+^ cells, 1.7 ± 0.2 vs. 0.0 ± 0.0; and F4/80^+^CD11b^+^ cells, 0.1 ± 0.0 vs. 0.0 ± 0.0, respectively.

### 3.3. Spleen Weights and the Proliferative Response of Splenocytes

Table 3 shows the spleen weights and the data of the proliferative responses by splenocytes on days 2, 4, and 14. On day 2, the spleen weights of the IL-18Rα KO mice were significantly lower than those of the WT mice. By days 4 and 14, there were no significant differences in spleen weight between the WT and IL-18Rα KO mice. On day 4, splenocytes obtained from IL-18Rα KO mice exhibited significantly less proliferation in the MTT assay compared to those from WT mice. On days 2 and 14, there were no significant differences in proliferation between the two groups.

### 3.4. Biomarkers of Arthritis

The levels of IL-6, IL-18, TNF, IFN-γ, and MMP-3 were measured as biomarkers in serum and culture supernatants (Figure 4). The serum IL-6 and IL-18 levels peaked at day 4 and the TNF and MMP-3 levels peaked at day 14 after LPS injection. Compared to the corresponding levels in the WT mice, these cytokines were significantly decreased in the IL-18Rα KO mice as follows. On day 4, the IL-6 levels were 225.0 ± 84.9 vs. 54.6 ± 44.8 pg/mL and the IL-18 levels were 274.8 ± 34.0 vs. 173.4 ± 45.2 pg/mL. The IFN-γ levels were 184.8 ± 55.7 vs. 84.5 ± 459.1 pg/mL on day 4 and 91.3 ± 60.5 vs. 32.5 ± 13.2 pg/mL on day 14. The TNF levels on day 2 were 2.3 ± 1.6 vs. 1.4 ± 0.8 pg/mL.

### 3.5. The Infiltration of CD4^+^ T Cells and Macrophages in the Inflamed Synovium

Figure 5 shows CD4^+^ T cells and F4/80^+^ cells in the synovium from a CIA mouse before and after an LPS injection. In the CIA mice without LPS injection, these cells were not detected in the synovium (Figure 5C,G). There were more of these cells in the IL-18Rα KO mice (Figure 5B,F) on days 4 and 14 after LPS injection compared to the corresponding numbers in the WT mice (Figure 5A,E).

We also evaluated the infiltration of CD4^+^ T cells and macrophages (i.e., inflammation cells) in the joint synovium after CIA. The infiltration of CD4^+^ T cells in WT mice increased to day 14 after LPS injection. The numbers of CD4^+^ T cells in the IL-18Rα KO mice were not increased at day 4 and were significantly decreased at day 14 after LPS injection (WT vs. IL-18Rα KO: 43.7 ± 8.9 vs. 7.8 ± 4.5 c/hpf, ** *p* < 0.01, Figure 5D). The numbers of macrophages (i.e., F4/80^+^ cells) in WT mice were increased and peaked at day 4, whereas the numbers of those cells in the IL-18Rα KO mice were decreased significantly compared to those in the WT mice at days 2 and 4 (WT vs. IL-18Rα KO: day 2, 44.0 ± 9.9 vs. 20.0 ± 3.9; day 4, 58.0 ± 7.1 vs. 19.0 ± 4.3 c/hpf; * *p* < 0.05, *** *p* < 0.001, Figure 5H).

### 3.6. Anti-Type II Collagen Antibody Levels

IgG, IgG1, IgG2a, and IgG2b antibody responses to type II collagen in WT and IL-18Rα KO mice were measured in the serum obtained on days 2, 4, and 14 (Figure 6). Compared to those in the WT mice, the levels of IgG2a and IgG2b anti-collagen antibodies were significantly lower in the IL-18Rα KO mice on day 4 (anti-collagen IgG2a: 635.0 ± 115.6 vs. 316.1 ± 83.7 units/mL; IgG2b: 106.0 ± 71.1 vs. 14.5 ± 45.2 Units/mL, respectively). In contrast, the levels of total IgG and IgG1 were not significantly different between the two groups of mice.

### 3.7. Inflammatory Cytokine and Biomarker mRNA Expressions in Inflamed Synovium

Inflammatory cytokines are well documented in CIA mice [20], and we thus measured the mRNA expressions in the synovium by conducting a real-time PCR (Table 4). In the IL-18Rα KO mice, there were widespread reductions in the mRNA expressions of cytokines (IL-6, IL-18, IFN-γ, and TNF) throughout the experimental course. On day 14, these cytokines were downregulated and showed significant differences between the WT and IL-18Rα KO groups.

We also assessed the expressions of mRNAs for two Th-cell subset transcription factors, T-bet (Th1) and GATA3 (Th2), in each mouse group. The T-bet expressions on day 14 after the LPS injection were reduced in the IL-18Rα KO mice compared to those in the WT mice, but the GATA3 expression was not changed in either group. Therefore, IL-18Rα deficiency shifted systemic responses away from the Th1 phenotype.

### 3.8. Intracellular TNFα and IFN-γ Staining in CD4^+^ T Cells and APCs after CIA-Induced Arthritis

We reported that the IL-18Rα signaling pathway has an important role in the response to TNFα and IFN-γ in CD4^+^ T cells among APCs [16]. In the present study, we also observed a strong reduction of the serum levels and mRNA expressions of TNFα and IFN-γ in inflamed synovium from the IL-18Rα KO mice. Table 5 provides the results of TNFα and IFN-γ^+^ staining of intracellular CD4^+^ T cells and the FACS analysis of F4/80^+^ cells and CD11b^+^ cells as APCs. On day 4, the IFN-γ^+^ staining in the CD4^+^ T cells, F4/80^+^ cells, CD11b^+^ cells, and double-positive F4/80^+^/CD11b^+^ cells in the IL-18Rα KO mice was significantly decreased compared to that in the WT mice. Similarly, the TNFα^+^ staining in the F4/80^+^ cells, the CD11b^+^ cells, and the double-positive F4/80^+^/CD11b^+^ cells in the IL-18Rα KO mice was significantly decreased compared to that in the WT mice, even though there was no significant between-group difference in CD4^+^ T cells.

### 3.9. Inflammatory Cytokine Production In Vitro

To investigate the role of IL-18/IL-18Rα in the production of proinflammatory cytokines, we stimulated splenocytes from CIA mice with ConA (1 μg/mL) and LPS (2 μg/mL) for 24 h. As shown in Figure 7, the stimulation with ConA and LPS induced the production of TNF and IL-6 as measured by ELISAs. Compared to the values in splenocytes from WT mice, the productions of TNF and IL-6 were reduced significantly in the splenocytes from IL-18Rα mice (TNF: 347.3 ± 27.9 vs. 209.5 ± 23.8 pg/mL; IL-6: 300.6 ± 18.2 vs. 173.0 ± 14.1 pg/mL, respectively). The IFN-γ and IL-18 levels were equivalent in the unstimulated and LPS-stimulated splenocytes, but compared to those from WT mice the IFN-γ levels were reduced in the splenocytes from IL-18Rα mice: 1478.0 ± 77.5 vs. 913.8 ± 57.0 pg/mL, respectively.

### 3.10. SOCS3 Expression in LPS-Induced CIA

Figure 8 illustrates the mRNA expression and the protein levels of splenic SOCS3 after LPS injection as shown by real-time PCR and Western blotting. On day 2, the mRNA expression of SOCS3 was significantly increased in the IL-18Rα KO mice compared to the WT mice: 8.9 ± 0.3 vs. 5.8 ± 0.6, respectively. The SOCS3 protein level also showed a trend of increase from day 2 to day 14 in the IL-18Rα KO mice, as follows: WT vs. IL-18Rα KO: day 2, 0.6 ± 0.2 vs. 0.3 ± 0.1; day 4, 0.5 ± 0.2 vs. 0.2 ± 0.1; day 14, 0.8 ± 0.1 vs. 0.2 ± 0.0. Western blotting and real-time PCR were performed but did not detect SOCS3 mRNA or protein in the splenocytes (data not shown).

## 4. Discussion

We have published several studies about IL-18Rα. In a mouse model of acute kidney injury (AKI), our findings indicated that IL-18Rα may mediate anti-inflammatory responses through SOCS1 and/or -3 in cisplatin-induced AKI [19]. Our later investigation showed that IL-18Rα also mediates apoptosis in a murine model of ischemia/reperfusion injury [21]. In a mouse model of lupus (autoimmune nephritis), we observed that MRL-Faslpr mice (well known as a lupus model) cross-bred with mice deficient in IL-18Rα had a better survival rate and lessened nephritis because of their reduced levels of autoantibodies [15]. However, it remains unknown how the IL-18/IL-18Rα signaling pathway functions in the immune response to autoimmune diseases such as rheumatoid arthritis. It is well known that CIA is accelerated by LPS, a major component of the outer membrane of gram-negative bacteria [22,23,24]. CIA after LPS injection is more useful than CIA alone for screening for anti-rheumatic drugs because it enables not only shortening the duration of experiments but also more accurate evaluation of incidence and severity of arthritis [22].

IL-18 appears to play an important role in the pathophysiology of experimental animal models of inflammatory arthritis. The incidence and severity of CIA in mice were worsened by the administration of IL-18 during the four-day period following initial and booster injections of collagen type II on days 0 and 21 [25]. These effects of IL-18 were due both to enhanced Th1 responses with increased IFN-γ production and to an IFN-γ-independent direct stimulation of TNFα production [26]. IL-18 was also suggested to be involved in rheumatoid arthritis, as IL-18 mRNA and protein were present in rheumatoid synovial tissue at higher levels than in osteoarthritis [25,27,28]. In addition, IL-18 enhanced the effects of IL-12 on the induction of IFN-γ production in synovial tissues in vitro and enhanced the stimulation of TNFα production in synovial fluid macrophages [25, 27, 28]. Given the known effects of IL-18 on stimulation of the production of proinflammatory cytokines, we suspect that IL-18 may be one of the factors responsible for the overproduction of TNFα and IL-1β in the rheumatoid synovium.

Banda et al. reported that the administration of murine IL-18 binding protein (mIL-18 BP) at the time of a booster injection of collagen type II reduced the progressive severity of CIA; both the clinical disease activity and histological scores were decreased by 50% [8]. Our present findings demonstrate that the numbers of inflammatory cells in the synovium were decreased and the severity of the arthritis score was ameliorated in IL-18Rα-deficient mice with LPS-induced arthritis. In another study, the productions of IFN-γ, TNFα, and IL-1β in LPS-stimulated splenocytes from C57BL6 mice were markedly decreased with mIL-18BP in vitro, and these decreased productions were concluded to be secondary to in vivo inhibitory effects on macrophages [8].

We have sought to determine the mechanisms underlying the activation of IL-18/IL-18Rα signaling in the immune system’s response to LPS-induced arthritis. To gain a better understanding of these issues, we investigated the effects of inflammatory cytokines that are accompanied by the influx into the synovium of monocytes such as APCs and CD4+ T cells in the acute advanced phase. Honda et al. showed that CD4+ T cells express surface T-cell receptors (TCRs) that identify foreign antigenic peptides bound to major histocompatibility complex (MHC) molecules on the surface of APCs in the LN [29]. These naive T cells are induced to proliferate rapidly by signaling via the TCRs and then integrate additional signals that allow them to differentiate into Th1, Th2, or Th17 phenotypes [30,31,32].

Of particular interest is the report that Th1 cells can gain the capacity to secrete IFN-γ (which is a key cytokine in the activation of macrophages and host protection against intracellular pathogens) and IL-18 [33,34]. After effector CD4+ T cells recognize peptide/MHC complexes displayed on tissue APCs [35,36], they can secrete cytokines, and the inflammatory cytokine production is thus targeted locally to the precise anatomical site of infection. Notably, IFN-γ and IL-18 elicited from Th1 cells can also activate neighboring macrophages [19,36], providing some nonspecific defense in the area around the Th1 stimulation. T-bet (*Tbx21*), a novel member of the T-box family of transcription factors, is a key regulator of Th1 lineage commitment and is required for optimal production of IFN-γ by CD4+ T cells [37]. It is primarily induced in response to IFN-γ/Stat1 signaling, up-regulating expression of both IFN-γ and the IL-12 receptor β2 chain (IL-12Rβ2), thereby enabling IL-12-induced stabilization of IFN-γ production and activation of the IL-18 signaling pathway. GATA3 is also essential for early thymocyte and mature peripheral T cell development [38]. In T cells, GATA3 is the master transcription factor driving naïve T cell differentiation to effector Th2 cells. Conversely, GATA3 suppresses Th1 and Th17 cell differentiation. The expression of mRNA for Th-cell subset transcription factors, T-bet (Th1) and GATA-3 (Th2) were assessed in mRNA from each mouse (Table 4). The T-bet expressions after the LPS injection were reduced in the IL-18Rα KO mice, but the GATA3 expression was not changed in either group. These results suggested that IL-18Rα deficiency shifted systemic responses away from the Th1 phenotype via IL-18 signaling pathway.

There are several cytokine-deficient mouse strains that are resistant to the lethal actions of LPS [36]. For example, Okamura et al. reported that IFN-γR KO mice displayed enhanced resistance to LPS, and the severity of the clinical changes caused by LPS (e.g., liver injury, weight loss) was greatly decreased in the mice [39]. However, IFN-γ production can itself be modulated by regulatory cytokines such as IL-12 and IL-18. IL-12 can both induce IFN-γ production and act synergistically with IL-18 [40]. IFN-γ production by B cells and macrophages occurred only after simultaneous treatment with IL-12 and IL-18 [35,41], whereas it was reported that T cells and natural killer (NK) cells do not require either IL-12 or IL-18 to produce IFN-γ [35,41]. Unusually high levels of IL-18 (which can induce IFN-γ production) and TNF-α have been described in humans [42]. Such IL-18 levels are associated with an imbalance between the natural inhibitor of IL-18 (i.e., IL-18BP) and IL-18 that is followed by an excess of free IL-18 [43]. The IL-18BP promoter has an element that is responsive to IFN-γ, and IFN-γ can effectively induce the production of IL-18BP [44].

In the present study, we investigated the effects of the expression of inflammatory cytokines (which is accompanied by an influx of monocytes including CD4+ T cells and APCs) in IL-18Rα KO mice with CIA by LPS injection. Our results demonstrated that the severity of arthritis was improved by suppressing serum inflammatory cytokines and the synovium cytokine mRNA expressions (especially those of IL-18, IFN-γ, TNF, and IL-6) and by suppressing the accumulation of CD4+ T cells in IL-18Rα KO mice. Specific cell-surface receptors mediate the binding of IL-18 to its target cells; this is similar to the IL-1R mechanism. An α-chain (IL-18Rα) [45,46] and a β-chain (IL-18Rβ or AcPL) comprise the receptor of IL-18 [47].

Born et al. proposed that IL-18Rβ does not itself interact directly with IL-18; they also suggested that IL-18Rα is responsible for the IL-18 binding. We investigated whether an IL-18Rβ signaling pathway contributes to decreased renal function. In a previous study, we treated WT mice with an anti-IL-18Rβ antibody in a cisplatin-induced renal injury model [19], and we found that this treatment reduced the renal function of the mice. The details of the involvement of an IL-18Rβ signaling pathway in the immune system remain unknown. Further studies are necessary to understand how IL-18Rβ affects immune responses.

In the acute phase in CIA produced by LPS injection, our present findings demonstrated that the expressions of IL-18 and IL-18Rα mRNA in CD4+ T cells and the numbers of IL-18R1+ cells among CD4+ T cells and CD11b+ cells from LN peaked at 4 h. Moreover, the severity of arthritis and bone erosions were ameliorated by suppressing the serum inflammatory cytokines and the synovium cytokine mRNA expressions (especially those of IL-18, IFN-γ, TNF, and IL-6) and by the suppression of an accumulation of CD4+ T cells and F4/80+ cells in IL-18Rα KO mice. To understand the pathogenesis underlying the immune response via the IL-18/IL-18Rα signaling pathway when mice are exposed to CIA, we evaluated the role of the SOCS pathway in this model.

The inhibition of cytokine signaling by the SOCS family constitutes a major negative feedback mechanism to prevent runaway inflammation. The transcription of SOCS proteins is rapidly upregulated in cells stimulated with pro-inflammatory cytokines. The SOCS then reduce the impact of cytokines by interacting with JAKs and by other mechanisms [9]. In addition, SOCS1 [10] and SOCS3 [14] interfere with IL-1/TLR-MAPK/NF-κB pathway signaling, and the increase in the activity of these kinases in IL-18Rα-deficient cells may be due to less inhibition resulting from reduced levels of both of these SOCS family members. It was reported that the mRNA expression and protein production of SOCS1 and SOCS3 were greatly reduced in cells from IL-18Rα-deficient mice compared to cells from WT mice [48].

The pathophysiology of rheumatoid arthritis (RA) characteristically involves several deregulated cellular events that are responsible for the altered innate and adaptive immune responses which significantly impact the structural alterations observed in articular cartilage and subchondral bone in this disease [49,50]. Swierkot et al. [51] indicated that the receptor activator of nuclear factor-κB ligand (RANKL) expression, which in RA contributes to the elevated level of osteoclast differentiation, is regulated by the IL-6/soluble IL-6 receptor, the JAK2/STAT3/SOCS-3 pathway. They also showed that the calcineurin inhibitor tacrolimus inhibited RANKL expression in RA-fibroblast-like synoviocytes by suppressing STAT3. Most importantly, Swierkot et al. showed that SOCS-3 was also induced by tacrolimus. Taken together, these results suggest that reduced SOCS-3 levels in RA may be responsible in part for initiating subchondral bone erosion. In addition, Shouda et al. [52] predicted that over-expressing SOCS3 in RA synovial tissue might provide a clinical benefit by dampening the clinical symptoms of RA. In that regard, Mahony et al. [53] proposed in their 2016 review that it may now be appropriate to focus narrowly on SOCS-3 as a therapeutic target in RA.

We observed a down-regulation of SOCS3 in IL-18Rα KO mice compared to wild-type mice. With an up-regulation of SOCS, pro-inflammatory cytokines were decreased through an IL-1/TLR-MAPK/NF-κB and/or JAK2/STAT3 signaling pathway. Our present findings indicate that IL-18Rα may induce, through SOCS3, anti-inflammatory responses in addition to pro-injury responses.

In summary, the up-regulation of SOCS was shown to decrease pro-inflammatory cytokines through the IL-18/IL-18Rα signaling pathway. These results suggest that blocking the IL-18/IL-18Rα signaling pathway could become a new treatment for rheumatoid arthritis.

## 5. Conclusions

The up-regulation of SOCS was shown to decrease pro-inflammatory cytokines through the IL-18/IL-18Rα signaling pathway. These results suggest that blocking the IL-18/IL-18Rα signaling pathway could become a new treatment for rheumatoid arthritis.

## Figures and Tables

**Figure 1 cells-09-00011-f001:**
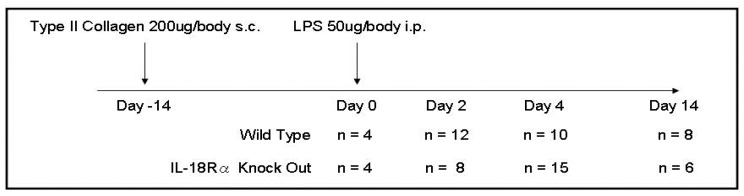
Experimental schedule in wild type and IL-18Rα knock-out mice for the induction of collagen-induced arthritis (CIA). The experimental set-up for CIA. Male DBA/1 mice at 8–10 weeks of age were immunized subcutaneously at the base of the tail with collagen type II and Freund’s adjuvant (day 14). On day 0, mice were injected with lipopolysaccharide (LPS) intraperitoneally for the acceleration of arthritis. Experiments were terminated at days 2, 4, and 14 after the first immunization. Day 2 was considered the (asymptomatic) induction phase of CIA, and days 4 and 14 were defined as the progress phase and established phase, respectively.

**Figure 2 cells-09-00011-f002:**
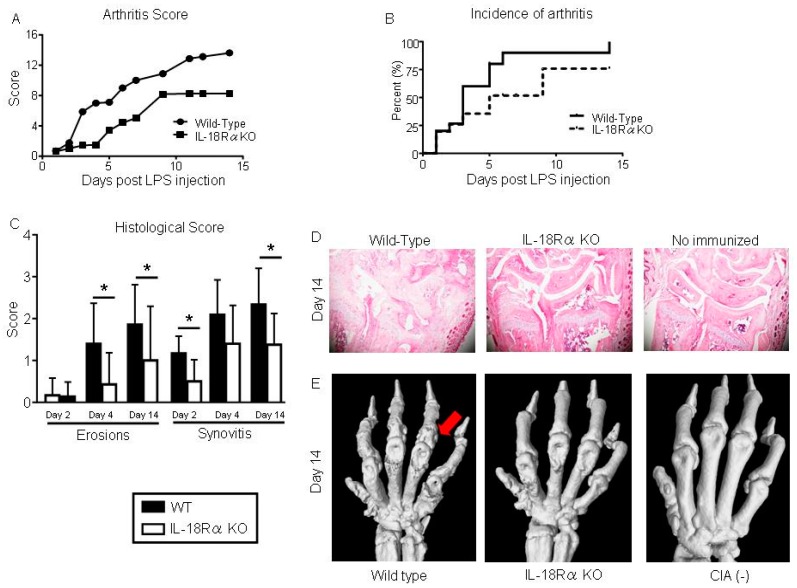
Inhibition of collagen-induced arthritis (CIA) in IL-18Rα knock-out (KO) mice. (**A**) Disease severity was assessed by visual inspection and scored on a scale of 0–4 per paw, 2×/week. The arthritis score (0–16) of CIA in the wild-type (WT; n = 30) and IL-18Rα KO mice (n = 29) after lipopolysaccharide (LPS) injection was calculated for each treatment group, and differences between groups were analyzed by the Mann-Whitney test. (**B**) The incidence of arthritis from grade 1 in each mouse is presented as Kaplan-Meier curves and was analyzed by log-rank test. (**C**) Microscopic synovitis and erosions on bone and cartilage were assessed the histological score in wrist sections stained with hematoxylin and eosin (H&E) from WT and IL-18Rα KO mice with CIA on day 2 (n = 12 and n = 8), 4 (n = 10 and n = 15), and 14 (n = 8 and n = 6). (**D**) Representative histological H&E images of mouse wrist joints in the WT, IL-18Rα KO, and non-immunized mice on day 14. (**E**) Visualization of CIA-induced joint damage in the wrist joints by micro-CT on day 14. Micro-CT imaging of a representative wrist from an IL-18Rα KO mouse with CIA shows normalized joint architecture and the prevention of bone destruction (*red arrow*). The data are mean ± SEM. * *p* < 0.05, ** *p* < 0.01, WT vs. IL-18Rα KO mice. Original magnification, ×40.

**Figure 3 cells-09-00011-f003:**
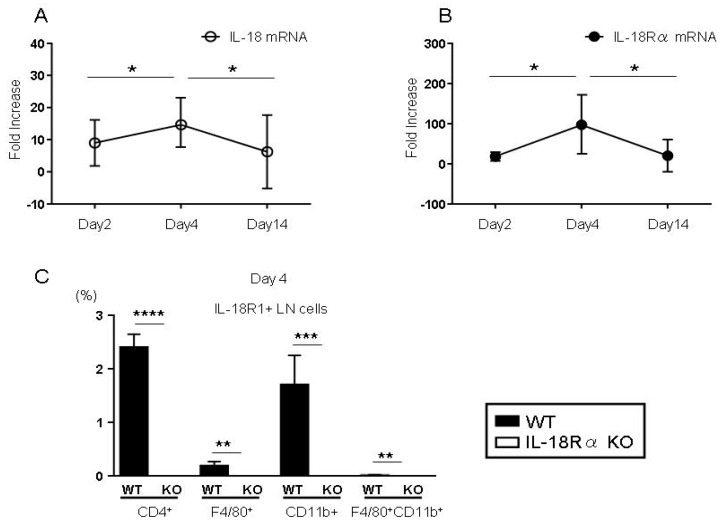
CIA affects the expression levels of IL-18 and IL-18Rα in the mouse synovium and lymph node (LN) cells. In the basic research, wild-type (WT) mice were immunized subcutaneously at the base of the tail with collagen type II and Freund’s adjuvant and injected intraperitoneally with 50 μg of lipopolysaccharide (LPS) and sacrificed on days 2 (n = 6), 4 (n = 6), and 14 (n = 6). The gene expressions of IL-18 (**A**) and IL-18Rα (**B**) in the synovium were measured by real-time PCR. In each experiment, the expression levels were normalized to the expression of 18SrRNA and are expressed relative to the values of saline-treated control mice. The data are the mean fold-increase and mean ± SEM: * *p* < 0.05, the WT mice after LPS injection at day 2 and 14 (n = 4 and n = 6) vs. day 4 (n = 4) after the LPS injection. The percentage of IL-18R1^+^ cells in CD4^+^ T cells, F4/80^+^, CD11b^+^ and F4/80^+^CD11b^+^ cells was measured by FACS analysis of LN cells on day 4 (**C**). The data are the mean ± SEM: ** *p* < 0.01, *** *p* < 0.005, and **** *p* < 0.001, WT vs. IL-18Rα KO mice on day 4 after the LPS injection (n = 10 and n = 15).

**Figure 4 cells-09-00011-f004:**
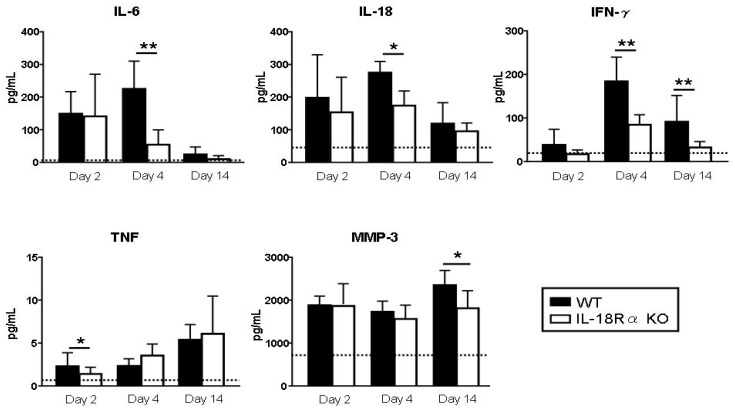
Effect of IL-18Rα on the production of inflammatory cytokines after collagen-induced arthritis (CIA). Serum tumor necrosis factor (TNF), interferon (IFN)-γ, IL-6, IL-12p40, IL-18, and matrix metalloproteinase-3 (MMP-3) productions were measured as biomarkers of arthritis on days 2 (n = 12 and n = 8), 4 (n = 10 and n = 15), and 14 (n = 8 and n = 6) in wild-type (WT) and IL-18Rα KO mice after lipopolysaccharide (LPS) injection. *Dotted lines* represent mean values from normal DBA1/J mice without bovine type II collagen and LPS injection. The data are mean ± SEM. * *p* < 0.05, ** *p* < 0.01, WT vs. IL-18Rα KO mice.

**Figure 5 cells-09-00011-f005:**
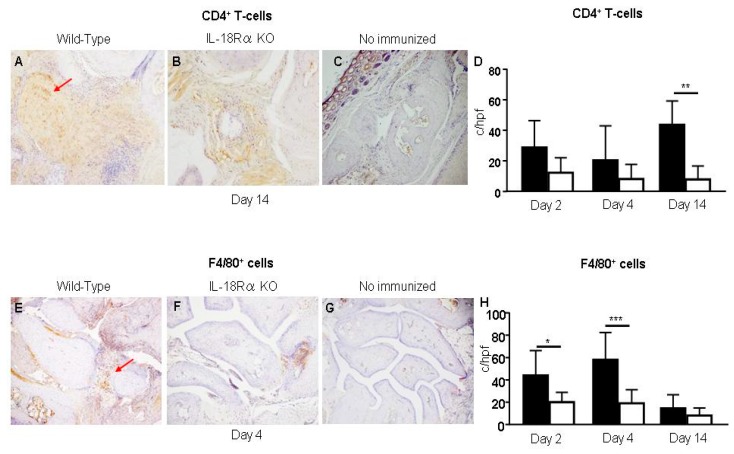
Effect of IL-18Rα on the accumulation of inflammation cells in the synovium after collagen-induced arthritis (CIA). No CD4^+^ T cells or F4/80^+^ cells were observed in the WT or IL-18Rα KO mice without immunization (**C**,**G**: n = 3). On days 4 and 14 after lipopolysaccharide (LPS) injection, there were more CD4^+^ T cells and F4/80^+^ cells (indicated by *red arrows* at high power, ×400) in the WT mice (**A**,**E**: n = 10) than in the IL-18Rα KO mice (**B**,**F**: n = 15). In the IL-18Rα KO mice, there were significantly fewer CD4^+^ T cells (**D**) and F4/80^+^ cells (**H**). Photomicrographs were taken at ×400. Values are mean ± SEM. * *p* < 0.05, ** *p* < 0.01, *** *p* < 0.005.

**Figure 6 cells-09-00011-f006:**
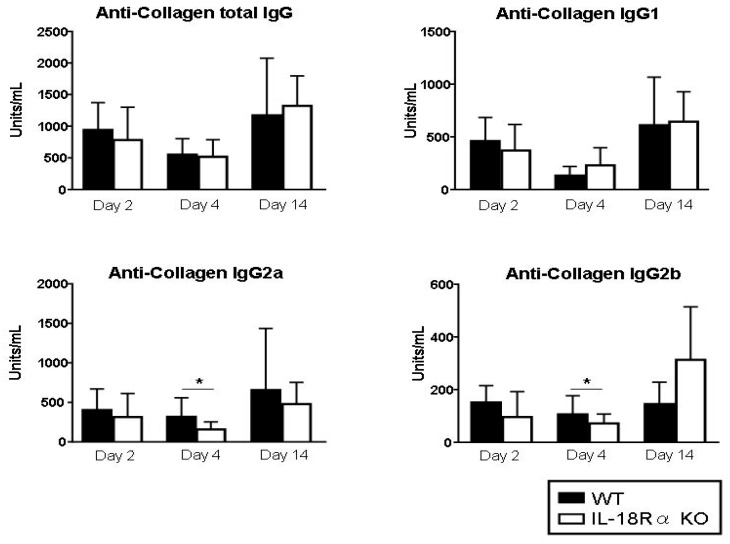
Effect of IL-18Rα on the production of anti-collagen antibodies after collagen-induced arthritis (CIA). Serum samples were collected from the wild-type (WT) and IL-18Rα KO mice on days 0 (n = 4 and n = 4), 2 (n = 12 and n = 8), 4 (n = 10 and n = 15), and 14 (n = 8 and n = 6). Each sample was analyzed by an ELISA. The levels of total IgG, IgG1, IgG2a, and IgG2b-anti-collagen antibodies in the WT and IL-18Rα KO mice and from the mice not injected with type II collagen were measured. Values are mean ± SEM. * *p* < 0.05.

**Figure 7 cells-09-00011-f007:**
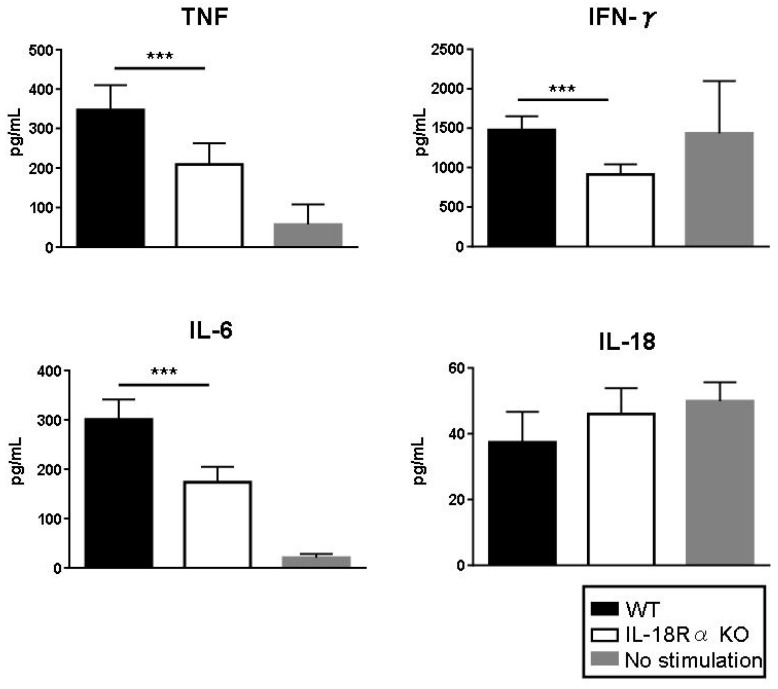
Role of IL-18/IL-18Rα signaling in proinflammatory cytokine production. To investigate the roles of IL-18/IL-18Rα on the production of proinflammatory cytokines, we used ConA (1 μg/mL) and lipopolysaccharide (LPS 2 μg/mL) in the presence of type II collagen (50 μg/mL) for 24 h to stimulate splenocytes in wild-type (WT) and IL-18Rα KO mice (n = 5 and n = 5). Stimulation with ConA and LPS induced the productions of tumor necrosis factor (TNF) and IL-6 as measured by ELISAs. Splenocytes from WT mice with no stimulation were used as controls (n = 5). The data are mean ± SEM. * *p* < 0.05, *** *p* < 0.001, WT vs. IL-18Rα KO mice.

**Figure 8 cells-09-00011-f008:**
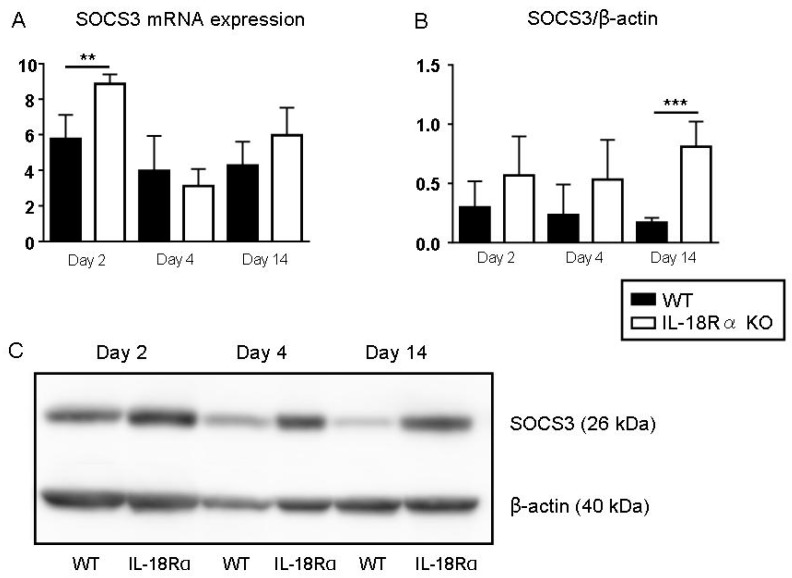
Effect of IL-18Rα on the suppressors of cytokine signaling (SOCS) expression after collagen-induced arthritis (CIA). The gene expressions (**A**) in wild-type (WT) and IL-18Rα KO mice on day 2 (n = 12 and 8), day 4 (n = 10 and 15), and day 14 (n = 8 and 6) after LPS injection were assayed by real-time PCR. Protein levels (**B**) were evaluated by Western blotting (WT and IL-18Rα KO, n = 5 each). A representative band of SOCS3 and β-actin is shown (**C**). The data are mean ± SEM. ** *p* < 0.01, *** *p* < 0.001, WT vs. IL-18Rα KO mice.

**Table 1 cells-09-00011-t001:** Primer sequences for the analysis of mRNA expression.

	Forward Primer	Reverse Primer
18SrRNA	GTAACCCGTTGAACCCCATTC	GCCTCACTAAACCATCCAATCG
IFN-γ	TGCTGATGGGAGGAGATGTCT	TTTCTTTCAGGGACAGCCTGTT
TNF	CGATCACCCCGAAGTTCAGTA	GGTGCCTATGTCTCAGCCTCTT
MMP-3	GGGAAGCTGGACTCCAACAC	GCGAACCTGGGAAGGTACTG
GATA3	AGGGACATCCTGCGCGAACTGT	CATCTTCCGGTTTCGGGTCTGG
T-bet	CCTGGACCCAACTGTCAACT	AACTGTGTTCCCGAGGTGTC

IL: interleukin; IFN: interferon; TNF: tumor necrosis factor; MMP-3: matrix metalloproteinase-3.

**Table 2 cells-09-00011-t002:** Gene database number for the analysis of mRNA expression.

	Gene Database No.
18SrRNA	NM_026744.3
IFN-γ	NM_008337.3
IL-6	Mm00446190
IL-18	NM_008360.1
IL-18R1 (IL-18Rα)	Mm00515180_m1

IL: interleukin; IFN-γ: interferon-gamma.

**Table 3 cells-09-00011-t003:** Effects of spleen weight and proliferative response on splenocytes in collagen-induced arthritis.

	Day 2	Day 4	Day 14
	WT vs. IL-18Rα KO	WT vs. IL-18Rα KO	WT vs. IL-18Rα KO
Spleen weight, g	0.2 ± 0.1 vs. 0.1 ± 0.1 *	0.1 ± 0.1 vs. 0.1 ± 0.0	0.1 ± 0.1 vs. 0.1 ± 0.0
MTT assay, O.D.	0.4 ± 0.2 vs. 0.4 ± 0.1	0.6 ± 0.1 vs. 0.5 ± 0.1 *	0.4 ± 0.2 vs. 0.4 ± 0.2

Spleen weights and MTT levels as parameters of splenic proliferation were measured in IL-18Rα KO and WT mice at days 2 (n = 8 and n = 12), 4 (n = 15 and n = 10), and 14 (n = 6 and n = 8) in CIA. The spleen cells (4 × 10^6^ cells/well) were stimulated with type II collagen in the presence of ConA (1 μg/mL) and LPS (2 μg/mL) for 48 h. The data are mean fold-increase ± SEM (* *p* < 0.05: IL-18Rα KO vs. WT mice). CIA: collagen-induced arthritis; WT: wild-type; IL-18Rα KO: interleukin-18 receptor α knock-out; LPS: lipopolysaccharide.

**Table 4 cells-09-00011-t004:** Effect of IL-18Rα on gene expression in CIA-induced arthritis.

Synovium	Day 2	Day 4	Day 14
	WT vs. IL-18Rα KO	WT vs. IL-18Rα KO	WT vs. IL-18Rα KO
Cytokines:			
IL-6	2677 ± 843.7 vs. 4.6 ± 4.1 **	348.5 ± 235.5 vs. 8.1 ± 7.3 **	42.4 ± 21.2 vs. 2.2 ± 2.2 **
IL-18	9.0 ± 7.2 vs. 0.1 ± 0.1 **	14.7 ± 6.8 vs. 0.3 ± 0.2 **	6.3 ± 5.7 vs. 1.0 ± 0.6 **
IFN-γ	32.3 ± 12.5 vs. 0.1 ± 0.2 **	137.0 ± 101.6 vs. 0.6 ± 0.8 *	23.2 ± 2.5 vs. 2.5 ± 1.4 *
TNF	2.4 ± 1.6 vs. 0.9 ± 0.7 *	2.4 ± 1.1 vs. 1.1 ± 1.2 *	0.3 ± 0.2 vs. 0.5 ± 0.1

MMP-3	25.7 ± 10.9 vs. 6.3 ± 4.8 *	49.3 ± 16.8 vs. 23.5 ± 14.1 *	25.1 ± 14.3 vs. 6.3 ± 4.8 *
Th-cell subset transcription factors:			
GATA3	0.1 ± 0.1 vs. 0.2 ± 0.1	6.1 ± 3.2 vs. 2.9 ± 0.8	1.3 ± 1.1 vs. 4.8 ± 4.5
T-bet	13.0 ± 3.4 vs. 14.3 ± 7.0	6.8 ± 2.1 vs. 6.2 ± 2.3	13.9 ± 35.0 vs. 3.7 ± 3.6

Cytokine, chemokine, and Th-cell subset transcription factor gene expressions were measured in IL-18Rα KO and WT mice at days 2 (n = 4 and n = 4), 4 (n = 15 and n = 10), and 14 (n = 6 and n = 8) after an LPS injection by real-time PCR. In each experiment, the expression levels were normalized to the expression of 18S rRNA and are expressed relative to the values of saline-treated control mice. The data are mean fold-increase ± SEM. * *p* < 0.05, ** *p* < 0.01: IL-18Rα KO vs. WT mice. CIA: collagen-induced arthritis; IFN: interferon; IL: interleukin; IL-18Rα KO: interleukin-18 receptor α knock-out; LPS: lipopolysaccharide; TNF: tumor necrosis factor; WT: wild type.

**Table 5 cells-09-00011-t005:** Intracellular TNFα and IFN-γ staining in CD4+ T cells and macrophages after CIA.

Day 4	WT vs. IL-18Rα KO	
	TNFα	IFN-γ
CD4^+^	3.8 ± 0.5 vs. 4.4 ± 0.6	1.4 ± 0.1 vs. 0.9 ± 0.1 **
F4/80^+^	2.3 ± 0.6 vs. 0.9 ± 0.1 *	1.5 ± 0.4 vs. 0.6 ± 0.1 *
CD11b^+^	3.0 ± 0.7 vs. 1.3 ± 0.1 *	2.1 ± 0.5 vs. 0.8 ± 0.1 *
F4/80^+^CD11b^+^	1.2 ± 0.7 vs. 0.5 ± 0.1 *	0.8 ± 0.5 vs. 0.3 ± 0.1 *

The percentages of TNFα^+^ and IFN-γ^+^ in CD4^+^ T cells, F4/80^+^, CD11b^+^, and F4/80^+^CD11b^+^ cells were measured by FACS analysis. The data are mean ± SEM. * *p* < 0.05 and ** *p* < 0.01, WT (n = 10) vs. IL-18Rα KO mice (n = 15). CIA: collagen-induced arthritis; IFN-γ: interferon-gamma; IL-18Rα KO: interleukin-18 receptor-alpha knock-out; LPS: lipopolysaccharide; TNF: tumor necrosis factor; WT: wild-type.

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
