# Peer review of "Inhibition of the IL-18 Receptor Signaling Pathway Ameliorates Disease in a Murine Model of Rheumatoid Arthritis"

_cells, 2019, doi:10.3390/cells9010011_

Round 1
Reviewer 1 Report
Dear Authors;
I have appreciated the manuscript submitted; I have found interesting the aim, the experimental design and adequate and coherent the discussion and the conclusions about the role of the IL-18 pathway in the mouse model of arthritis discussed.
However there are some issues that I advice to reconsider and eventually change for the next draft.
1) It's not clearly exposed the number of mice used for the part of the experiment (point 2.2). The partial numbers (for example n=4 at day zero...n=10 and 15 at day 4 should be , a priori, described from a sample size process. I suggest to re-organize the chapter in order to a better understanding of the design and to ameliorate the statistic comprehension.
2) The CIA mouse model used is intriguing even really rarely used. In figure 1 (line 94) , showing the experimental timeline you write about mRNA expression of IL18 and receptor. But it seem to be an incongruence within the text.
3) Chapter 2.5 describes the use of Micro-CT , however , beside a qualitative consideration there aren't other results obtained with this technique. It should be nice if the manuscript shows some quantitative data (especially about erosion in the joints bones affected). In my opinion this could support the decision to use an experimental model that is enhanced by LPS respect to the classical models of CIA.
4) The chapter 3.1 should be post-poned to chapter 3.2 for a better coherence of the discussions.
5) I suggest , if it possible to describe where , anatomically (axillary, neck region, mediasthinic...) have been extracted the limph nodes used in the experiments.
6) In chapter 3.7 (line 318) there is t world " kidney" but probably is a mistake.
7) In the manuscript you should, in my opionion , describe more extensively the GATA3 and T-Bet pathway in order to show the biological role in the frame of the experimental model shown.
8) Emphatizing the role of Lps in arthropathies you should discuss and cite some works about the links between Lps or other gram - toxins and joint diseases as rheumatoid arthritis. This could enrich the translational value of this paper.
In conclusion i have appreciated the study however i consider the suggestions above to be necessary to do the manuscirpt noteworthy completely.
Best regards.
Author Response
We would like to thank the referee for his/her helpful comments, as well as the positive comments on our studies and the manuscript, and we have amended the descriptions in response to his/her suggestions.
Dear Authors;
I have appreciated the manuscript submitted; I have found interesting the aim, the experimental design and adequate and coherent the discussion and the conclusions about the role of the IL-18 pathway in the mouse model of arthritis discussed.
However there are some issues that I advice to reconsider and eventually change for the next draft.
☑It's not clearly exposed the number of mice used for the part of the experiment (point 2.2). The partial numbers (for example n=4 at day zero...n=10 and 15 at day 4 should be , a priori, described from a sample size process. I suggest to re-organize the chapter in order to a better understanding of the design and to ameliorate the statistic comprehension.
Response
Thank you for your suggestion. Like you pointed, the present description was very confusing. We inserted description the sample numbers (WT and KO mice) in all Figure and Table legend and added the description about the sample numbers in the section Statistical analyses 2.11. If there is still confusing, please point again.
Line 200-203
In vivo, the sample data were analyzed from WT and IL-18Rα KO mice; day 0 (n=4 and n=4), 2 (n=12 and n=8), 4 (n=10 and n=15), and 14 (n=8 and n=6) in all figures except the sample data in Figure 3A on day 2, 4, 14 (n=6). In vitro, the splenocytes were extracted from the WT, IL-18Rα KO, and control mice (n=5) in Figure 7.
List of Figure and Tables
Figure 2; Line 224, 229, and 232
Figure 3; Line 253 and 257
Figure 4; Line 287-288
Figure 5; Line 308-310
Figure 6; Line 321-322
Figure 7; Line 375 and 377
Figure 8; Line 389
Table 3; Line 272
Table 4; Line 338
Table 5; Line 357-358
☑ The CIA mouse model used is intriguing even really rarely used. In figure 1 (line 94), showing the experimental timeline you write about mRNA expression of IL18 and receptor. But it seem to be an incongruence within the text.
Response
Thank you for your pointing out. It seems to be incongruence with the experimental schedule and mRNA data in Figure 3 (Figure 2 previously).We delete the description about mRNA expression and wrote the new legend title in Figure 1.
Line 94-95
Figure 1. Experimental schedule in wild type and IL-18Rα knock-out mice for the induction of collagen-induced arthritis (CIA).
We are also sorry to describe the wrong sentence in Figure 3 (Figure 2 previously). We amended the description in the accordance with Figure 1.
Line 256
the WT mice after LPS injection at day 2 and 14 (n=4 and n=6) vs. day 4 (n=4) after the LPS injection. The percentage of IL-18R1+ cells in CD4+ T cells, F4/80+, CD11b+ and F4/80+CD11b+ cells was measured by FACS analysis of lymphoid (LN) cells on day 4 (C). The data are the mean ± SEM: **p < 0.01, p***<0.005, and p****<0.001, WT vs. IL-18Rα KO mice on day 4 after the LPS injection (n=10 and n=15).
We mentioned about the LPS can accelerate the arthritis in this study to shorten the duration of experiments and accurate evaluation in the incidence and severity of arthritis.
Line 402-405
It is well known that CIA is accelerated by LPS, a major component of the outer membrane of gram‐negative bacteria [22-24]. CIA after LPS injection is more useful than CIA alone for screening for anti-rheumatic drugs because it enables not only shortening the duration of experiments but also more accurate evaluation of incidence and severity of arthritis [22].
☑ Chapter 2.5 describes the use of Micro-CT, however , beside a qualitative consideration there aren't other results obtained with this technique. It should be nice if the manuscript shows some quantitative data (especially about erosion in the joints bones affected). In my opinion this could support the decision to use an experimental model that is enhanced by LPS respect to the classical models of CIA.
Response
Thank you for your suggestion. Like your suggestion, we tried to measure the volume in erosive area of bone be Micro-CT, but the image was not worth of evaluation it for the quantitative data. Thank you for your suggestion again, and we would like to keep only the image without quantification in this study.
☑ The chapter 3.1 should be post-poned to chapter 3.2 for a better coherence of the discussions.
Response
Thank you for pointing this out. We post-poned the chapter 1 after chapter 3.2 and swapped the Figure 2 for 3
.
Line 206-260
☑ I suggest, if it possible to describe where, anatomically (axillary, neck region, mediasthinic...) have been extracted the limph nodes used in the experiments.
Response
Thank you for pointing this out. We amended the description from where lymph nodes were extracted.
Line 174
For the assessment of intracellular cytokines by a FACSCanto II flow cytometry system (Becton Dickinson, Lincoln Park, NJ), we obtained lymphoid cells from axillary and neck region at days 2, 4, and 14 and splenocytes at day 4 after LPS injection
☑ In chapter 3.7 (line 318) there is t world " kidney" but probably is a mistake.
Response
Thank you for pointing this out. We amended from the kidney to the synovium.
Line 327
Inflammatory cytokines and chemokines are well documented in CIA mice [20], and we thus measured the mRNA expressions in the synovium by conducting a real-time PCR (Table 4).
☑ In the manuscript you should, in my opionion , describe more extensively the GATA3 and T-Bet pathway in order to show the biological role in the frame of the experimental model shown.
Response
Thank you for pointing this out. We described and inserted the adequate references [38 and 39] the GATA3 (Th2) and T-Bet pathway in the text.
Line 441-453
T-bet (Tbx21), a novel member of the T-box family of transcription factors, is a key regulator of Th1 lineage commitment and is required for optimal production of IFN-γ by CD4+ T cells [38]. It is primarily induced in response to IFN-γ/Stat1 signaling, up-regulating expression of both IFN-γ and the IL-12 receptor β2 chain (IL-12Rβ2), thereby enabling IL-12-induced stabilization of IFN-γ production and activation of the IL-18 signaling pathway. GATA3 is also essential for early thymocyte and mature peripheral T cell development [39]. In T cells, GATA3 is the master transcription factor driving naïve T cell differentiation to effector Th2 cells. Conversely, GATA3 suppresses Th1 and Th17 cell differentiation. Expression of mRNA for Th-cell subset transcription factors, T-bet (Th1) and GATA-3 (Th2) were assessed in mRNA from each mouse (Table 4). The T-bet expressions after the LPS injection were reduced in the IL-18Rα KO mice, but the GATA3 expression was not changed in either group. These results suggested that IL-18Rα deficiency shifted systemic responses away from the Th1 phenotype via IL-18 signaling pathway.
☑ Emphatizing the role of Lps in arthropathies you should discuss and cite some works about the links between Lps or other gram - toxins and joint diseases as rheumatoid arthritis. This could enrich the translational value of this paper.
Response
Thank you for pointing this out. We described that the LPS is closely correlates with autoimmune arthritis in the text.
Line 402-405
It is well known that CIA is accelerated by LPS, a major component of the outer membrane of gram‐negative bacteria [22-24]. CIA after LPS injection is more useful than CIA alone for screening for anti-rheumatic drugs because it enables not only shortening the duration of experiments but also more accurate evaluation of incidence and severity of arthritis [22].
In conclusion i have appreciated the study however i consider the suggestions above to be necessary to do the manuscirpt noteworthy completely.
Best regards
Reviewer 2 Report
In this manuscript, Nozaki Y & al demonstrated that the inhibition of the IL-18 receptor signalling pathway ameliorates disease in a murine model of RA. They found that the numbers of inflammatory cells in the synovium were decreased and that the severity of the arthritis was improved in IL-18Rα-deficient mice in whom arthritis were induced by administration of LPS. They also found that the improvement of arthritis in IL-18Rα-deficient mice correlated with lower levels of pro-inflammatory cytokines in the serum and reduced accumulation of CD4+ T cells in the synovium of mice. This protective effect seems to be linked to an increase of SOC3 expression in IL-18Rα-deficient mice, as demonstrated by quantitative RT-PCR and western blot experiments.
Before publication, the following points should be considered:
In the Result part:
Line 211 to 216: Please add a reference to the Figure 2.
In the legend of the Figure 2: detail the parts of the Figure (A, B, C)
In the diagram of the Figure 2C, the legend of the axis is only: %. Specify what percentage it is.
Line 317: The authors referred to results of chemokine mRNA expression, but these results do not appear in any figure?? It seems that the authors have quantified mRNA of CCL2/MCP-1, and the results are expected to be present in the Table 4, but it is not the case….
Line 372: Reference to Figure 7 is not correct. The results exposed in this paragraph are those that are shown in Figure 8 (I suppose that it is Figure 8 because in the legend of the figure of the page 13 of the manuscript, the number of the Figure is absent!!)
Moreover, in the legend of this figure (page 13), the authors have written that: “Protein levels (B) were evaluated by Western blotting and immunohistochemistry (WT and IL-18Rα KO, n=5 each)”, but I do not see any result of immunochemistry...
Author Response
We would like to thank the referee for his/her helpful comments, as well as the positive comments on our studies and the manuscript, and we have amended the descriptions in response to his/her suggestions.
In this manuscript, Nozaki Y & al demonstrated that the inhibition of the IL-18 receptor signalling pathway ameliorates disease in a murine model of RA. They found that the numbers of inflammatory cells in the synovium were decreased and that the severity of the arthritis was improved in IL-18Rα-deficient mice in whom arthritis were induced by administration of LPS. They also found that the improvement of arthritis in IL-18Rα-deficient mice correlated with lower levels of pro-inflammatory cytokines in the serum and reduced accumulation of CD4+ T cells in the synovium of mice. This protective effect seems to be linked to an increase of SOC3 expression in IL-18Rα-deficient mice, as demonstrated by quantitative RT-PCR and western blot experiments.
Before publication, the following points should be considered:
In the Result part:
☑ Line 211 to 216: Please add a reference to the Figure 2.
Response
Thank you for pointing this out. We added the reference to the Figure 2 (the current Figure 3) in the text.
Line 243-245
We also examined the expressions of IL-18R1+ on CD4+ T cells, F4/80+ cells, CD11b+ cells, and F4/80+CD11b+ cells by performing a FACS analysis of lymphoid cells in WT and IL-18Rα KO mice on day 4 (Fig. 3C).
☑ In the legend of the Figure 2 (the current Figure 3): detail the parts of the Figure (A, B, C)
Response
Thank you for pointing this out. We added the sentence about the Figure 3C and described about the results in FACS analysis in the legend.
Line 250-260
Figure 3. CIA affects the expression levels of IL-18 and IL-18Rα in the mouse synovium and LN cells. In the basic research, wild-type (WT) mice were immunized subcutaneously at the base of the tail with collagen type II and Freund's adjuvant and injected intraperitoneally with 50 μg of LPS and sacrificed on days 2 (n=6), 4 (n=6), and 14 (n=6). The gene expressions of IL-18 (A) and IL-18Rα (B) in the synovium were measured by real-time PCR. In each experiment, the expression levels were normalized to the expression of 18SrRNA and are expressed relative to the values of saline-treated control mice. The data are the mean fold-increase and mean ± SEM: *p<0.05, the WT mice after LPS injection at day 2 and 14 (n=4 and n=6) vs. day 4 (n=4) after the LPS injection. The percentage of IL-18R1+ cells in CD4+ T cells, F4/80+, CD11b+ and F4/80+CD11b+ cells was measured by FACS analysis of lymphoid (LN) cells on day 4 (C). The data are the mean ± SEM: **p < 0.01, p***<0.005, and p****<0.001, WT vs. IL-18Rα KO mice on day 4 after the LPS injection (n=10 and n=15).
☑In the diagram of the Figure 2C, the legend of the axis is only: %. Specify what percentage it is.
Response
Thank you for pointing this out. The axis is the percentages of IL-18R1+ cells in CD4+ T cells, F4/80+, CD11b+ and F4/80+CD11b+ cells was measured by FACS analysis of lymphoid (LN) cells. We added the sentence to specify the axis for what percentage it is about the Figure 2C in the legend.
Line 257-259
The percentage of IL-18R1+ cells in CD4+ T cells, F4/80+, CD11b+ and F4/80+CD11b+ cells was measured by FACS analysis of lymphoid (LN) cells on day 4 (C).
☑ Line 317: The authors referred to results of chemokine mRNA expression, but these results do not appear in any figure?? It seems that the authors have quantified mRNA of CCL2/MCP-1, and the results are expected to be present in the Table 4, but it is not the case….
Response
Thank you for pointing. We are sorry to describe the wrong sentence. In this study, we omitted the results in hemokine mRNA expression for CCL2/MCP-1: chemokine (C-C motif) ligand2/monocyte chemoattractant protein-1. We amended the sentences as follow;
Line 326-330
Inflammatory cytokines are well documented in CIA mice [20], and we thus measured the mRNA expressions in the synovium by conducting a real-time PCR (Table 4). In the IL-18Rα KO mice, there were widespread reductions in the cytokines mRNA expressions (IL-6, IL-18, IFN-γ, and TNF) throughout the experimental course. On day 14, these cytokines were downregulated and showed significant differences between the WT and IL-18Rα KO groups.
☑ Line 372: Reference to Figure 7 is not correct. The results exposed in this paragraph are those that are shown in Figure 8 (I suppose that it is Figure 8 because in the legend of the figure of the page 13 of the manuscript, the number of the Figure is absent!!)
Response
Thank you for pointing this out. We are sorry to describe the wrong Figure number. Figure 8 is correct in the results and we added the title in Figure 8.
Line 380-381
Figure 8 illustrates the mRNA expression and the protein levels of splenic SOCS3 after LPS injection as shown by real-time PCR and Western blotting.
Line 388
Figure 8. Effect of IL-18Rα on the SOCS expression after CIA
☑ Moreover, in the legend of this figure (page 13), the authors have written that: “Protein levels (B) were evaluated by Western blotting and immunohistochemistry (WT and IL-18Rα KO, n=5 each)”, but I do not see any result of immunochemistry...
Response
Thank you for pointing this out. We are sorry to describe the wrong sentence and deleted the word of immunohistochemistry
Line 390; deleted the sentence as the below
and immunohistochemistry
Round 2
Reviewer 2 Report
The authors have satisfactorily responded to all my questions and made the necessary changes to the manuscript.